# Bioprinted Cancer Model of Neuroblastoma in a Renal Microenvironment as an Efficiently Applicable Drug Testing Platform

**DOI:** 10.3390/ijms23010122

**Published:** 2021-12-23

**Authors:** Dongwei Wu, Johanna Berg, Birte Arlt, Viola Röhrs, Munir A. Al-Zeer, Hedwig E. Deubzer, Jens Kurreck

**Affiliations:** 1Institute of Biotechnology, Chair of Applied Biochemistry, Technische Universität Berlin, 13355 Berlin, Germany; dongwei.wu@campus.tu-berlin.de (D.W.); johanna.berg@tu-berlin.de (J.B.); viola.roehrs@tu-berlin.de (V.R.); al-zeer@tu-berlin.de (M.A.A.-Z.); 2Department of Pediatric Hematology and Oncology, Charité-Universitätsmedizin Berlin, 13353 Berlin, Germany; birte.arlt@charite.de (B.A.); hedwig.deubzer@charite.de (H.E.D.); 3Neuroblastoma Research Group, Experimental and Clinical Research Center (ECRC) of the Charité and the Max-Delbrück-Center for Molecular Medicine (MDC) in the Helmholtz Association, 13125 Berlin, Germany; 4German Cancer Consortium (Deutsches Konsortium für Translationale Krebsforschung, DKTK), Partner Site Berlin, 10115 Berlin, Germany; 5Berliner Institut für Gesundheitsforschung (BIH), 10178 Berlin, Germany

**Keywords:** bioprinting, cancer model, drug testing, neuroblastoma, panobinostat

## Abstract

Development of new anticancer drugs with currently available animal models is hampered by the fact that human cancer cells are embedded in an animal-derived environment. Neuroblastoma is the most common extracranial solid malignancy of childhood. Major obstacles include managing chemotherapy-resistant relapses and resistance to induction therapy, leading to early death in very-high-risk patients. Here, we present a three-dimensional (3D) model for neuroblastoma composed of IMR-32 cells with amplified genes of the *myelocytomatosis viral related oncogene* *MYCN* and the *anaplastic lymphoma kinase* (*ALK*) in a renal environment of exclusively human origin, made of human embryonic kidney 293 cells and primary human kidney fibroblasts. The model was produced with two pneumatic extrusion printheads using a commercially available bioprinter. Two drugs were exemplarily tested in this model: While the histone deacetylase inhibitor panobinostat selectively killed the cancer cells by apoptosis induction but did not affect renal cells in the therapeutically effective concentration range, the peptidyl nucleoside antibiotic blasticidin induced cell death in both cell types. Importantly, differences in sensitivity between two-dimensional (2D) and 3D cultures were cell-type specific, making the therapeutic window broader in the bioprinted model and demonstrating the value of studying anticancer drugs in human 3D models. Altogether, this cancer model allows testing cytotoxicity and tumor selectivity of new anticancer drugs, and the open scaffold design enables the free exchange of tumor and microenvironment by any cell type.

## 1. Introduction

Bioprinting has been attracting a great deal of attention as a promising technology to produce three-dimensional (3D) tissue models [1,2,3,4]. It allows the production of 3D constructs with high spatial resolution by successively adding material in a layer-by-layer manner. Most commonly, cell-laden hydrogels are used as bioinks that are rapidly crosslinked after the printing procedure to maintain the desired structure [5]. With optimized hydrogel compositions, bioprinted cultures can be maintained for an extended period of time while retaining high cell viability for the duration of the experiment [6].

Bioprinting technology is particularly suitable for the creation of tumor models, as the high precision and reproducibility can recapitulate the tumor microenvironment (TME) [7,8]. In most animal models, human tumors are embedded in a xenogenic animal environment [9]. This arrangement may produce results with limited relevance to human pathophysiology. Bioprinting may help to overcome this shortcoming [10,11]. For example, Langer et al. modeled tumor phenotypes, in which patient-specific tumor tissue was surrounded by several stromal cell types [12]. Extrinsic signals and therapies altered the tumor phenotypes, and the printed model was used to investigate interaction between cancer cells and their microenvironment, demonstrating the potential of bioprinting for the development of new anticancer drugs.

The present study focuses exemplarily on neuroblastoma, which is the most common extracranial solid tumor of childhood that derives from developing and incompletely committed precursor cells from neural-crest tissues [13,14,15]. Despite progress in the treatment and the use of multi-modal therapy, survival rates of high-risk neuroblastoma patients are still low [16]. Dysregulation of the transcription factor MYCN is associated with poor prognosis [17]. Amplification of this proto-oncogene acts as a single oncogenic driver towards high-risk neoplastic transformation [18]. Panobinostat is a potent histone deacetylase (HDAC) inhibitor approved by the U.S. Food and Drug Administration (FDA) for the treatment of multiple myeloma and is currently under investigation against various other cancer types [19]. As an additional important factor, forkhead-box-protein O3 (FOXO3) was found to be an important regulator of homeostasis that promotes tumor growth under hypoxic conditions and tumor angiogenesis in late-stage neuroblastoma [20]. Further targets in high-risk neuroblastoma include the telomerase reverse transcriptase (TERT) and the oncogene ALK [21,22]. Phosphoglycerate dehydrogenase (PHGDH) is a suitable marker for risk stratification, as it is highly upregulated in high-risk *MYCN*-amplified neuroblastoma; however, its inhibition by small molecule inhibitors antagonized chemotherapy efficiency in patient-derived xenografts in mice [23]. In a recent study by Almstedt et al., 80 targets were found to be associated with the risk of neuroblastoma, and differentiation signatures and candidates for the treatment of high-risk neuroblastoma were identified [24].

Neuroblastomas have a high potential to migrate and can metastasize to almost any organ. Around 60% of patients with neuroblastoma develop metastases, most commonly involving bone marrow or cortical bone [25]. Although renal metastasis from neuroblastoma is rather rare, cases have been reported [26,27,28]. Especially for bilateral renal metastases or multiple renal metastases, local therapeutic options for the kidneys, such as nephrectomy and/or radiotherapy, are infeasible, as they can cause complete loss of renal function in patients [29]. Over the years, little improvement in the treatment for neuroblastoma renal metastasis has been obtained and progress in understanding of the disease and the development of new therapeutic strategies are urgently awaited.

The aim of the present study was to develop a bioprinted 3D model that mimics a tumor in a microenvironment exclusively composed of human cells. As neuroblastoma cells have been shown to be well suited for bioprinting approaches [30,31,32,33,34,35], this tumor type was chosen as an example. To the best of our knowledge, few studies exist that have embedded the tumor cells in an environment of normal cells to test the efficiency and specificity of cytostatic or other anticancer drugs. Our study describes the creation of a renal neuroblastoma model, in which the neuroblastoma cells were surrounded by a microenvironment made up of human kidney cells. It can thus be regarded as a simplified metastasis model. The model was created with a commercially available printer to allow simple reproduction by other groups. We demonstrate that it can distinguish between cancer-specific drugs and substances with general cytotoxicity and can thus be used for the development of new cancer drugs or personalized treatment strategies. It can also be seen as a model to reflect neuroblastoma infiltration into the kidney, as this process presents a major medical problem, and if patients are poor responders to chemotherapy, nephrectomy can be indicated.

## 2. Results

### 2.1. Drug Treatment of Mono-Cell Type 3D Culture

The first step in the development of a cancer model was to characterize the individual drug-sensitivity of the employed cell types. For the initial experiments, the neuroblastoma cell line IMR-32 and the human embryonic kidney 293 cells (HEK293) were printed into simple 3D grid-like structure (Figure 1a) in a gelatin-alginate bioink as previously described [36].

As a proof-of-concept of the bioprinted cancer model for use in drug testing, the constructs were treated with varying concentrations of the cancer drug panobinostat one day after the printing procedure. Relative cell viabilities were determined with XTT assays (2,3-Bis-(2-Methoxy-4-Nitro-5-Sulfophenyl)-2H-Tetrazolium-5-Carboxanilide) after 24, 48, and 72 h. Dose–response curves show a significantly lower sensitivity of HEK293 cells towards panobinostat treatment compared to IMR-32 cells (Figure 2a,b). While the IC_50_ values of IMR-32 cells were in the low nanomolar range, they were in the range of hundreds of nanomolar for HEK293 cells (Figure 2b,d and Table 1).

For comparison, we tested the effects of blasticidin, which is an unspecific antibiotic substance that inhibits translation [37]. As can be seen in Figure 2c, dose–response curves of HEK293 and IMR-32 cells were similar for blasticidin. IC_50_ values of both cell types were comparable and did not have significant differences at the time points under investigation (Figure 2d and Table 1).

### 2.2. Cytotoxicity in 3D Constructs

Cytotoxicity can be directly monitored in 3D constructs to assess the cytostatic impact caused by panobinostat on bioprinted cells. HEK293 and IMR-32 cells were separately printed in grid models and treated with varying concentrations of panobinostat. After 72 h, the ratio of live (green channel) and dead (red channel) cells in the constructs was monitored by fluorescence microscopy using a cytotoxicity assay (Figure 3a,b). Percentages of live and dead cells, resulting from quantification of green and red fluorescence signals, were calculated by the software ImageJ (Figure 3c,d). More than 75% of printed HEK293 cells survived using panobinostat concentrations of up to 50 nM, and only the highest doses led to an obvious increase of dead cells (Figure 3a,c). In contrast, panobinostat began cell killing at much lower doses and resulted in death of almost all IMR-32 cells already at a concentration of 10 nM and above (Figure 3b,d).

### 2.3. Cell Sensitivity in 2D Culture

To figure out the influence of the 3D arrangement of the cells, we compared the IC_50_ values obtained in the bioprinted models with those from 2D monolayer cultures. The 2D cultures were challenged with a single dose of either panobinostat at varying concentrations and cultured for 72 h. During the culture period, relative cell viability was monitored by XTT assays (Figure 4) and used to calculate the IC_50_ values from the dose–response curves (Table 1). As observed for the 3D cultures, IMR-32 cells were substantially more sensitive to panobinostat treatment than HEK293 cells, and viability was significantly decreased at concentrations in the low nanomolar range starting as early as 24 h post treatment. After 48 and 72 h of cultivation, the decrease in viability became even more pronounced and at concentrations above 15 nM of panobinostat, virtually no viable cells were detected. Accordingly, IC_50_ values of IMR-32 cells challenged with panobinostat were in the low nanomolar range and substantially lower than that of HEK293 cells (Table 1).

These results were confirmed by cytotoxicity assays (Appendix A), which clearly displayed differences in sensitivity of HEK293 and IMR-32 cells towards panobinostat treatment. While HEK293 cells were virtually insensitive to the panobinostat treatment in the concentration range tested, the fraction of green fluorescence from viable IMR-32 cells drastically decreased at panobinostat concentrations above 5 nM.

The most interesting outcome of the comparison was that the IC_50_ values of IMR-32 cells for panobinostat were approximately one order of magnitude higher for the 3D cultures than for the 2D monolayers. The differences were less pronounced for HEK293 cells so that the therapeutic window was broader in the bioprinted constructs, i.e., the difference in sensitivity between both cell types was more pronounced in 3D culture (approximately two orders of magnitude) than in 2D culture (roughly one order of magnitude).

As the study intended to investigate the specificity of treatment for cancerous cells, we also tested whether co-cultivation of both cell types in 2D influences the sensitivity towards panobinostat. To this end, HEK293 cells stably expressing green fluorescence protein (HEK293-GFP) and IMR-32 cells were seeded together at a ratio of 1:1. After treatment with panobinostat, cells were analyzed by an immunofluorescence microscopy (Figure 5). The green fluorescence emitted by HEK293-GFP cells was used to simplify this analysis. IMR-32 cells were labeled by immunofluorescence staining against human disialoganglioside GD2 (GD2, red channel), which is expressed on tumors of neuroectodermal origin, including neuroblastoma and melanoma [38,39]. Nuclear counterstaining was performed with DAPI (4′,6-diamidin-2-phenylindol, blue channel). As shown in Figure 5, HEK293-GFP and IMR-32 cells occupied approximately equivalent areas in the untreated control group after 72 h. With increasing panobinostat doses, the area with red signals, which represents the GD2-stained IMR-32 cells, shrank gradually, whereas green fluorescing HEK293 cells occupied the vacated area. Only at the highest concentration of panobinostat tested (50 nM), was a decrease in HEK293-GFP cells observed, while virtually no more IMR-32 cells were detectable.

Similar to the 3D constructs, the sensitivity of the cells in 2D monolayers for blasticidin was also tested. As previously observed, dose–response curves and calculated IC_50_ values were similar for both cell types and did not show significant differences at the time points under investigation (Figure 4c,d and Table 1).

### 2.4. Induction of Apoptosis in 2D Culture

Panobinostat is known to be an HDAC inhibitor, so our next aim was to confirm observation of this mode of action in our experimental set-up. This activity may result in the induction of apoptosis. We therefore investigated whether panobinostat treatment of HEK293 and IMR-32 cells in 2D culture produced cleaved caspase-3 (green channel in Appendix A) by immunofluorescence staining. Additionally, cellular filamentous actin (F-actin, red channel) and nuclei (blue channel) were visualized by phalloidin and DAPI counterstaining, respectively. Staining of F-actin and the nuclei revealed that increasing panobinostat concentrations led to decreasing numbers of IMR-32 cells but did not impact HEK293 cell numbers, which was in agreement with the results of XTT and cytotoxicity assays. Cleaved caspase-3 was not detected in HEK293 cells at panobinostat concentrations below 25 nM, and even 50 nM panobinostat resulted in only weak signals. In contrast, signals resulting from cleaved caspase-3 were detected in IMR-32 cells, even at concentrations as low as 5 nM, and became more pronounced at higher concentrations in a dose-dependent manner. This demonstrates that panobinostat is a stronger inducer of apoptosis in IMR-32 cells than in HEK293 cells.

### 2.5. Bioprinting and Drug Treatment of Cancer Model

After the initial characterization of the drug activity, a cancer model was fabricated that consisted of a cancerous core (IMR-32 cells) surrounded by a shell of kidney cells as illustrated in Figure 1b. In the initial experiments, HEK293-GFP cells were used for better visualization, then HEK293 cells were included to provide additional immunofluorescence evidence and in the final experiments primary kidney fibroblasts were used to increase the physiological significance of the model. The diameter of the inner core was 3 mm, while the total model was 6 mm in diameter and 0.4 mm in height (Figure 6a). A set of 48 such constructs were produced by bioprinting and proved to be highly reproducible (Figure 6b).

Fluorescence analyses revealed a clear boundary between the IMR-32 cell in the center and the green fluorescing HEK293-GFP cells (Figure 6c). The model was treated with panobinostat for 72 h and dead cells were stained with ethidium homodimer-1. Pronounced red fluorescence of dead cells was detected coming from the inner part composed of IMR-32 cells, while strong green fluorescence in the outer ring resulted from high GFP expression of the stably transfected HEK293-GFP cells. Only at very high panobinostat concentrations (1000 nM) was a fraction of dead, red fluorescent HEK293-GFP cells observed. The significantly higher sensitivity of the neuroblastoma cells towards panobinostat was clearly confirmed in the quantitative analysis of the red fluorescence of the inner and outer part of the model, respectively (Figure 6d). Two conclusions can be drawn from these observations: The bioink composition allows maintenance of the intended design of the model with a cancerous core surrounded by a shell of kidney cells, and the differences in drug sensitivity can be clearly seen in a 3D model composed of different cell types.

The experiments were repeated with HEK293 and IMR-32 cells, as this approach allows the detection of living cells by calcein AM staining, which cannot be distinguished from the green fluorescence of HEK293-GFP cells. These experiments confirmed the observations made above (Appendix A). A concentration of panobinostat as low as 10 nM was sufficient to kill a substantial fraction of IMR-32 cells. The merged images show a clear border between the dead, red fluorescing cells in the center and green living cells in the outer ring at panobinostat concentrations of 10 to 100 nM. Red fluorescence originating from dead HEK293 cells was only observed at very high panobinostat concentrations.

The next experiment aimed at investigating the induction of apoptosis in the different parts of the cancer model by immunofluorescent labeling of cleaved caspase-3 (Appendix A, green channel). To clearly distinguish HEK293 cells from IMR-32 cells in the printed cancer models, the latter were labeled with the neuroblastoma-specific GD2 antibody (red channel). Nuclear counterstaining with DAPI (blue channel) revealed a homogenous distribution of the cells throughout the constructs for all samples (Appendix A). Starting at a panobinostat concentration of 10 nM, green fluorescence indicating the induction of apoptosis became visible in the cancerous core of the constructs. Signal intensity increased at higher drug concentrations. In contrast, even at the highest concentration of panobinostat of 1000 nM, only a weak green signal originating from cleaved caspase-3 was observed in the periphery of the model containing HEK293 cells. Quantification of the fluorescence signals using ImageJ revealed significant differences between the presence of cleaved caspase-3 in the cancer part and surrounding renal environment (Appendix A). Thus, induction of apoptosis is significantly stronger by a factor of two to three in IMR-32 cells compared to that in HEK293 cells.

A completely different picture arose for blasticidin treatment. Here, no fluorescence originating from cleaved caspase-3 was observed at concentrations up to 10 µM (Appendix A). At higher concentrations, the signal became stronger in a dose-dependent manner in both parts of the model. Quantitative analysis of the fluorescence by ImageJ confirmed the blasticidin-sensitivity of both cell types is roughly equivalent (Appendix A). The model thus allows distinguishing drugs which are specifically toxic to cancer cells like panobinostat from those that are generally cytotoxic such as blasticidin.

### 2.6. Cell Response of Primary Human Kidney Fibroblasts on Panobinostat Treatment

Despite the ambiguities about the origin of the HEK293 cell line and its derivatives, they are among the most widely used cells in molecular biology, after HeLa cells [40]. To improve the (patho-)physiological relevance of the bioprinted cancer model, we replaced the HEK293 cells with human primary kidney fibroblasts, expecting them to provide a physiologically more relevant human renal microenvironment for the cancer cells. Fibroblasts are important regulators for the maintenance of tissue cohesion, as they are essential for the production and degradation of extracellular matrix components [41]. In addition, kidney fibroblasts also have endocrine activity [42].

For an initial characterization of their drug sensitivity, human kidney fibroblasts were seeded in a 96-well plate and treated with panobinostat. Cell viability, as evaluated by XTT assay, remained above 90% 24 h post treatment for all concentrations tested (Figure 7a). Only at later time points (48 and 72 h after panobinostat treatment), was a dose-dependent decrease in viability detected. The IC_50_ values were calculated and found to be comparable to the ones of HEK293 cells in 2D culture (Figure 7b and Table 2). Similar characteristics were observed when primary human kidney fibroblasts were printed into a 3D structure (Figure 7c,d). Compared to the IC_50_ values of the printed IMR-32 cells (see above, Table 1), IC_50_ values for the primary fibroblasts were approximately two orders of magnitude higher (Table 2). Resistance of human kidney fibroblasts to panobinostat was also found in cytotoxicity assay for 2D and 3D cultures (Appendix A).

### 2.7. Effect of Panobinostat on Printed Cancer Model with Neuroblastoma and Primary Kidney Fibroblasts

In the final experiment of this study, primary human kidney fibroblasts were printed in the cancer model described above, i.e., the center containing IMR-32 neuroblastoma cells was surrounded by a ring of primary fibroblasts. The model was treated with increasing concentrations of panobinostat, and cytotoxicity assays were carried out 72 h thereafter. As can be seen in Figure 8, red fluorescence originating from dead cells appeared in the cancerous part starting at 10 nM panobinostat and became more intense at increased drug concentrations. In contrast, dead fibroblasts were only observed at high concentrations of panobinostat.

## 3. Discussion

Despite substantial progress in the last few decades, efficient treatment options are still lacking for many tumor types and especially for metastatic cancer. In preclinical studies, 2D monolayer cultures have greatly contributed to the basic knowledge of genetic factors that drive transformation of somatic cells into tumor cells. These studies, however, cannot mimic the 3D architecture of tumors and their interaction with their surrounding microenvironment. To this end, animal models were developed which allowed studying tumor development in a complex pathophysiological environment [9]. Although these models made an enormous contribution to the field, they provide a xenogenic microenvironment instead of a human one, and therefore often have limited relevance to human pathophysiology [43]. As a consequence, the average success rate for the translation of insights from animal models to clinical trials is less than 8% [44]. In line with these data, a comprehensive review revealed a failure rate of drug candidates in oncology of 97% [45]. Alternative strategies with higher predictivity for newly developed drug candidates are thus urgently required.

Although still being a comparatively young discipline, bioprinting technologies have already demonstrated their potential for cancer research [7] and the printability of neuroblastoma cells has been demonstrated in several studies: The Noguera group produced bioprinted neuroblastoma models and investigated the impact of tissue stiffness, which commonly increases in solid tumors [30,31]. Remarkably, they found stiffness to influence expression patterns and cellular physiology. In a model composed of the neuroblastoma cell line SH-SY5Y in co-culture with mesenchymal stromal cells and human primary umbilical vein endothelial cells, the neuroblastoma cells formed Homer Wright-like rosettes and maintained their proliferative capacities [34]. Another bioprinted tumor model consisting of SK-N-BE(2) cells was used to investigate the infiltration of chimeric antigen receptor (CAR) T cells into tumor tissues [35]. In further studies, neuroblastoma cell lines have been used to develop neural tissue for studying neurodegenerative diseases [32,33]. None of these previous studies, however, tested cytostatic or other anticancer drugs in the bioprinted models. As the printability of neuroblastoma cells has been well documented, we chose this tumor type as an example for our open design of a cancer model to test the activity and specificity of anticancer drugs. Our model was produced with a commercially available bioprinter and can therefore easily be adapted by other research groups.

The TME has a major influence on the solid tumor, as it provides cytokines, immune cells, and vasculature that determine the tumor phenotype and encumber therapeutic interventions [46,47]. Due to species differences, the effects of the TME measured in animal models cannot be relied upon when translated into clinical settings. In contrast, bioprinting can be used to produce a tumor in a human TME and to investigate interactions between the tumor and its TME, as well as its influence on drug treatment, for neuroblastomas and other types of tumors [48]. In breast cancer, tumor progression is strongly influenced by its microenvironment and particularly by interaction of the cancer cells with adipose tissue, which can be recapitulated in bioprinted models [49]. In another study, breast cancer cells were printed in the center of a 3D model and surrounded by adipose-derived mesenchymal stem/stromal cells (ADMSC) [50]. This model was significantly less sensitive to treatment with doxorubicin than a construct that contained the cancer cells only. The response was found to depend on the thickness of the ADMSC layer, demonstrating the importance of the tumor environment. In a previous study, we found a bioprinted liver model to be less sensitive toward Aflatoxin B1 than a monolayer culture [51]. The possibility of creating sustainable long-term cultures allows the study of the long-term mutagenic effects of a potential carcinogen. Heterogenous tissue models with high cell density can be produced by bioprinting technologies, including spheroids [52].

The current lack of standards for models and their reproducibility makes it difficult to compare the results from different research groups. For example, Langer et al. produced a tumor model, as described above, in which cancer cells were printed in a stromal mix of human fibroblasts and endothelial cells [12]. This model studied interactions between the tumor and its microenvironment; however, the sophisticated model was produced with a special printer of the company Organovo, Inc. to which other researchers do not have access. In contrast, the model presented here can easily be reproduced with an affordable, commercially available printer.

In our study, we evaluated the sensitivity of the neuroblastoma cells IMR-32 and renal HEK293 cells, as well as primary kidney fibroblasts, toward the anticancer drug panobinostat and the cytotoxic substance blasticidin in 2D and 3D cultures. IMR-32 cells had comparable sensitivities for panobinostat in 2D and 3D cultures, whereas HEK293 and primary kidney fibroblasts became more resistant, when cultured in the 3D model. Most importantly, IMR-32 cells were substantially more sensitive to panobinostat treatment than the renal cells. The difference was approximately one order of magnitude in 2D culture and increased to roughly two orders in magnitude in bioprinted models. In contrast, the effect of blasticidin treatment was comparable for IMR-32 and HEK293 cells and the IC_50_ values increased for both cell types in 3D compared to 2D in a similar manner.

The bioprinting technology was then used to produce neuroblastoma in a renal environment. Fluorescence microscopy confirmed that the chosen bioinks maintained the intended structure over the course of the experiments for 72 h. Cytotoxicity assays showed that intermediate panobinostat concentrations of 10–100 nM selectively killed neuroblastoma cells, while leaving the kidney cells intact. Cell death occurred via the induction of apoptosis, as demonstrated by measuring increased levels of cleaved caspase-3. In contrast to panobinostat, blasticidin induced apoptosis in both cell types at similar concentrations.

As HEK293 cells are easy to culture and expand to large numbers, they were used for the initial experiments. This cell line is widely used, but its exact origin is still controversial [40]. While they have been considered kidney epithelial cells or fibroblasts, their karyotype is unstable, and they are tumorigenic. We therefore used primary kidney fibroblasts in further experiments. These tests confirmed findings obtained with HEK293 cells. IMR-32 cells are approximately two orders of magnitude more sensitive to panobinostat treatment in the cancer model than the kidney cells and can thus be selectively killed by the anti-tumor drug.

As described above, the study of Grundwald et al. [35] used a bioprinted neuroblastoma model consisting of SK-N-BE(2) cells to investigate tumor tissue penetration by CAR T cells. Our model, which consists of not only cancerous cells, but also normal fibroblasts, can now be used to investigate the specificity of the treatment for the destruction of the tumor. The next step will therefore be to use the model consisting of neuroblastoma in a microenvironment composed of non-cancerous cells, not only for cytostatic substances but also for immunotherapeutic approaches. Another interesting option is to use patient-derived tumor cells to develop a personalized treatment strategy. For example, Mao et al. produced a 3D tumor model with patient derived intrahepatic cholangiocarcinoma cells [53], and Flores–Torres et al. developed a patient-derived 3D bioprinted spheroid model with triple-negative breast cancer cells [54]. These studies, however, used the cancer cells only, while the strategy presented here will allow to study the patient-derived cells in a human TME.

While we focused the present study on the production of an advanced cancer model involving a tumor surrounded by a human microenvironment, at the same time we aimed to support the principles of replacement, reduction, and refinement (3R principles). Highly sophisticated 3D organ models will help to replace animal experiments with in vitro studies [55]. It is our strong belief that only innovative new tissue engineering strategies enabling better transferability of research results to the human (patho-)physiology will bring us closer to the ultimate goal to reduce the number of animals used for experimental purposes.

## 4. Materials and Methods

### 4.1. Cell Culture

Human embryonic kidney 293 cells (HEK293, CRL-1573) were purchased from American Type Culture Collection (ATCC, Manassas, VA, USA) and the neuroblastoma cell line IMR-32 from the German Collection of Microorganisms and Cell Cultures (DSMZ, Braunschweig, Germany). HEK293-GFP cells were obtained from GenTarget (SC001, San Diego, CA, USA). Human kidney fibroblasts were purchased from Innoprot (P10666, Derio, Bizkaia, Spain). All cell lines were cultured in Dulbecco’s modified Eagle’s medium (DMEM, Biowest, Nuaillé, France) containing 10% fetal bovine serum (FBS; c.c.pro, Oberdorla, Germany), 2.5 mg/mL of glucose (Biowest, Nuaillé, France), 2 mM of L-glutamine (Biowest, Nuaillé, France), 1% non-essential amino acids (NEAA, Biowest, Nuaillé, France), and 1% penicillin-streptomycin (Biowest, Nuaillé, France) and maintained at 37 °C in humidified atmosphere with 5% CO_2_. When confluent, the cells were washed with phosphate buffered saline (PBS, Biowest, Nuaillé, France) and then harvested using trypsin-EDTA (Biowest, Nuaillé, France).

### 4.2. Bioprinting

The hydrogel, consisting of 6.67% gelatin (Sigma–Aldrich, St. Louis, MO, USA) and 4.5% sodium alginate (Sigma, Shanghai, China), was prepared in DMEM under continuous stirring at 37 °C overnight as described before [36,56]. Prior to the printing process, the printable cell-laden bioink was obtained by mixing the hydrogel, CaSO_4_ (Roth, Karlsruhe, Germany), and the cell suspension, so that the final concentration of each component was: 3% gelatin, 2% sodium alginate, 30 mM of CaSO_4_, and 5 × 10^6^ cells/mL bioink. After physical pre-crosslinking for 8 min at room temperature, the cell-laden bioink was transferred into a pneumatic cartridge.

The 3D constructs were fabricated in a 48-well plate using a multi nozzle bioprinting system (Bio X, Cellink, Gothenburg, Sweden) as the bioink was extruded from a 22 G conical tip under pneumatic pressure. A double-layer grid-like model with a side length of 8 mm was printed for single-cell type printing, while a concentric disc construct with a 3 mm-diameter inner part containing cancer cells, and 6 mm-diameter outer part containing normal stromal cells was fabricated for the two-cell type cancer model. After the printing process, printed models were submerged in 100 mM of CaCl_2_ for 10 min at room temperature. Afterwards, the 100 mM CaCl_2_ solution was replaced with 300 µL of complete medium supplemented with 20 mM CaCl_2_ per well, and subsequently the constructs were cultured at 37 °C and 5% CO_2_.

### 4.3. Drug Treatment of Cancer Models

For monolayer culture, cells were seeded into a collagen (90 µg/mL, collagen type I, rat tail, EMD Millipore, Billerica, MA, USA) -coated 96-well plate at a density of 10^4^ cells/well. After culture for 24 h, the supernatant of each well was replaced by 100 µL of medium with the respective drug (panobinostat ((LBH589, Selleckchem, Houston, TX, USA), initially dissolved in dimethyl sulfoxide (DMSO, Sigma–Aldrich, St. Louis, MO, USA)) or blasticidin (10 mg/mL in 20 mM HEPES, Sigma–Aldrich, St. Louis, MO, USA)) at the indicated concentrations. Complete medium was used for the untreated control group.

For the 3D bioprinted constructs, complete medium was supplemented with 20 mM of CaCl_2_ and drugs at the given concentration range were used to treat the samples. The constructs cultured in complete medium supplemented with only 20 mM of CaCl_2_ served as untreated group.

### 4.4. Cell Viability Assay

Cell viability of the cultures was determined by XTT assays (2,3-Bis-(2-Methoxy-4-Nitro-5-Sulfophenyl)-2H-Tetrazolium-5-Carboxanilide, Alfa Aesar, Ward Hill, MA, USA) that measured metabolization of the tetrazolium salt at various time points following the treatment. Briefly, a mixture of 50 µL XTT reagent (1 mg/mL in RPMI, Biowest, Nuaillé, France) and phenazine methosulfate (PMS, 3.83 mg/mL in PBS, AppliChem, Darmstadt, Germany) at a volume ratio of 500:1 was added to each well of a 96-well plate for 2D cell culture and allowed to incubate for 4 h at 37 °C and 5% CO_2_. The absorbance was measured at wavelengths of 450 and 620 nm (for reference) using a microplate reader (Sunrise, Tecan, Männedorf, Switzerland). For 3D constructs, 150 µL of XTT/PMS reagent mixture was added for 4 h, and the absorbance of the supernatant was measured as mentioned above. Cell-free constructs were used as a background. The relative cell viability was calculated by the following formula:(1)$$\mathrm{relative}\;\mathrm{cell}\;\mathrm{viability}=\frac{{\mathrm{Absorbance}}_{\mathrm{test}\;\mathrm{well}}-{\mathrm{Absorbance}}_{\mathrm{background}\;\mathrm{well}}}{{\mathrm{Absorbance}}_{\mathrm{control}\;\mathrm{well}}-{\mathrm{Absorbance}}_{\mathrm{background}\;\mathrm{well}}}$$

Afterwards, half maximal inhibitory concentration (IC_50_) values were calculated based on the nonlinear regression curves of the dose–response data (dose–response curves) using GraphPad Prism 8 (GraphPad, La Jolla, CA, USA). All experiments were performed at least three times.

### 4.5. Cytotoxicity Assay

To analyze the cell status and cell distribution, a cytotoxicity assay was performed using a viability/cytotoxicity kit (Thermo Fisher Scientific, Waltham, MA, USA) in accordance with the manufacturer’s instructions. The 2D cultured cells were incubated in RPMI without phenol red, which contained 2 µM of calcein AM and 2 µM of ethidium homodimer-1 for 10 min, while the 3D constructs were incubated for 30 min. The stained samples were analyzed by fluorescence microscopy (Observer Z1, Zeiss, Jena, Germany). The ratio of living and dead cells in 3D printed constructs was also analyzed using the software ImageJ (1.53e, National Institutes of Health, Bethesda, MD, USA).

### 4.6. Immunofluorescence Staining

At predetermined time points, samples were washed with PBS and fixed in 4% formaldehyde (Carl Roth, Karlsruhe, Germany) for 30 (2D) or 60 min (3D) at room temperature. The samples were then permeabilized with 0.1% (*v*/*v*) Triton X-100 (Carl Roth, Karlsruhe, Germany) for 10 (2D) or 30 min (3D), and blocked with 5% goat serum (Sigma-Aldrich, St. Louis, MO, USA) for 30 (2D) or 60 min (3D), respectively. Afterwards, the samples were incubated with primary antibodies (anti-human disialoganglioside GD2, 1:400, BD Pharmingen, Franklin Lakes, NJ, USA; Cleaved Caspase-3 antibody, 1:1000, Cell Signaling, Danvers, MA, USA) at 4 °C overnight, washed three times with PBS, and subsequently incubated with the respective secondary antibodies (goat anti-mouse Alexa Fluor 594, 1:1000, Invitrogen, Carlsbad, CA, USA; goat anti-rabbit Alexa Fluor 488, 1:1000, Invitrogen, Carlsbad, CA, USA) at room temperature for 2 h. Afterwards, samples were again washed with PBS three times prior to nuclear staining with 1 µg/mL of 4′,6-diamidino-2-phenylindole (DAPI, Sigma–Aldrich, St. Louis, MO, USA) for 60 min. When indicated, F-actin of cells was labeled with phalloidin (Alexa Fluor™ 488 Phalloidin, 1:400, Invitrogen, Carlsbad, CA, USA) for 30 min at room temperature. Stained samples were analyzed by the fluorescence microscopy.

### 4.7. Statistical Analysis

Results are shown as the means ± standard error of the mean from at least three independent experiments. Statistical analyses were performed using GraphPad Prism 8 software. One-way ANOVA was utilized for analysis of variance to compare between groups. Statistical significance was accepted at levels of * *p* < 0.05, ** *p* < 0.01, *** *p* < 0.001, **** *p* < 0.0001.

## 5. Conclusions

Taken together, we present a neuroblastoma model that can easily be adapted to other cancer types as it allows replacing the cancer and surrounding cells with any cell type of interest. It may also be used to test tumor cells from a specific patient and develop a personalized treatment strategy. Two main conclusions can be drawn from our study: Bioprinted tumor models composed of cancerous cells in a non-malignant environment composed of human cells can be used to differentiate substances with a specific anticancer activity from those with general cytotoxic properties, and the sensitivity of cells towards cytotoxic substances differs substantially in 2D and 3D culture.

## Figures and Tables

**Figure 1 ijms-23-00122-f001:**
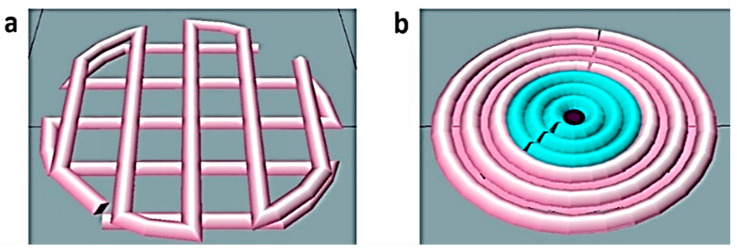
Schematic representation of the 3D printed constructs designed for the present study. For the initial experiments a simple grid-like structure was used (**a**), while the cancer model (**b**) consisted of a core of neuroblastoma cells (cyan) surrounded by healthy kidney cells (pink).

**Figure 2 ijms-23-00122-f002:**
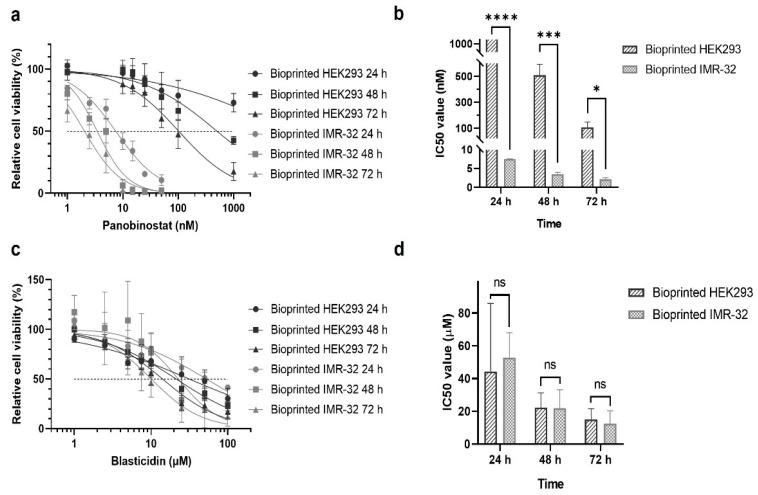
Sensitivity of 3D cultured HEK293 and IMR-32 cells towards panobinostat and blasticidin. HEK293 and IMR-32 cells were printed in a gelatin-alginate hydrogel (5 × 10^6^ cells/mL) in 48-well plates and treated with either panobinostat or blasticidin. Dose–response curves of bioprinted HEK293 and IMR-32 cells after treatment with panobinostat (**a**) or blasticidin (**c**) for 24, 48, and 72 h are shown. The calculated IC_50_ values for both cell types are compared for panobinostat in (**b**) and for blasticidin in (**d**). Data are presented as mean ± standard error of the mean; *n* = 3. * *p* < 0.05, *** *p* < 0.001, **** *p* < 0.0001.

**Figure 3 ijms-23-00122-f003:**
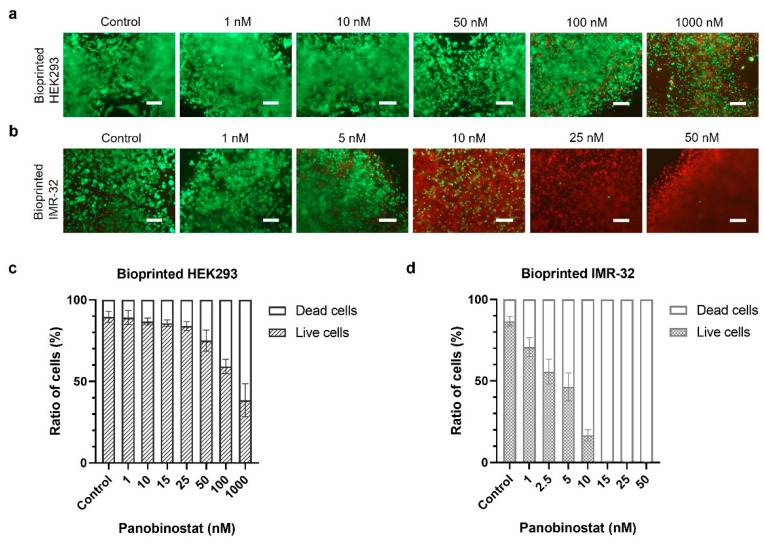
Cytotoxicity of HEK293 cells and IMR-32 cells in printed constructs after treatment with panobinostat. Cytotoxicity assays of 3D-bioprinted HEK293 cells (**a**) and IMR-32 cells (**b**) after treatment with panobinostat for 72 h (living cells in green, dead cells in red; all images were taken at the same magnification, scale bar, 250 μm), and the estimated percentages of living and dead cells (**c**,**d**).

**Figure 4 ijms-23-00122-f004:**
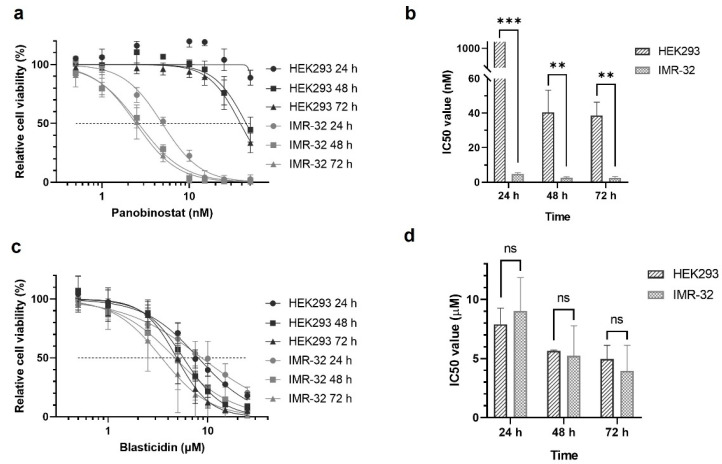
Sensitivity of 2D cultured HEK293 and IMR-32 cells towards panobinostat and blasticidin. HEK293 (10^4^ cells/well) and IMR-32 cells (10^4^ cells/well) were separately seeded in 96-well plates and treated with either panobinostat or blasticidin. Dose–response curves of HEK293 and IMR-32 cells after treatment with panobinostat (**a**) or blasticidin (**c**) 24, 48, and 72 h post treatment; and the calculated IC_50_ values of HEK293 and IMR-32 cells for indicated time points, (**b**) for panobinostat and (**d**) for blasticidin. Data are presented as mean ± standard error of the mean; *n* = 3. ** *p* < 0.01, *** *p* < 0.001.

**Figure 5 ijms-23-00122-f005:**
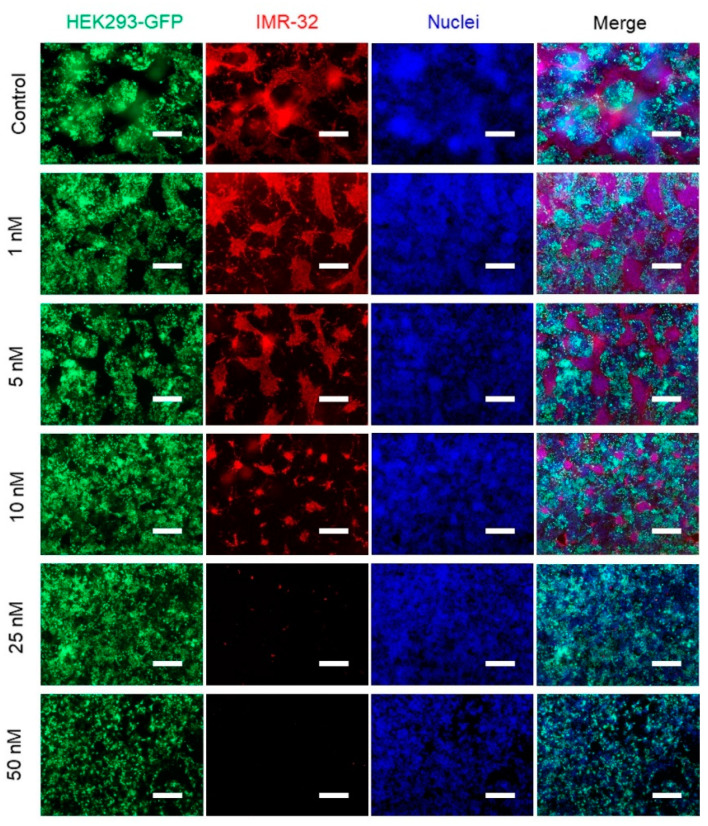
Co-culture of HEK29-GFP cells and IMR-32 cells in monolayer treated with panobinostat visualized by expressed GFP and immunofluorescence staining of GD2. HEK293-GFP cells and IMR-32 cells (1:1) at a total density of 10^4^/well were seeded in 96-well plate and dosed with panobinostat at the indicated concentrations for 72 h. HEK293-GFP cells (green) expressed GFP while IMR-32 (red) cells were labeled by GD2 immunofluorescence staining. The nuclei were all stained with DAPI in blue. All images were taken at the same magnification; scale bar, 250 μm.

**Figure 6 ijms-23-00122-f006:**
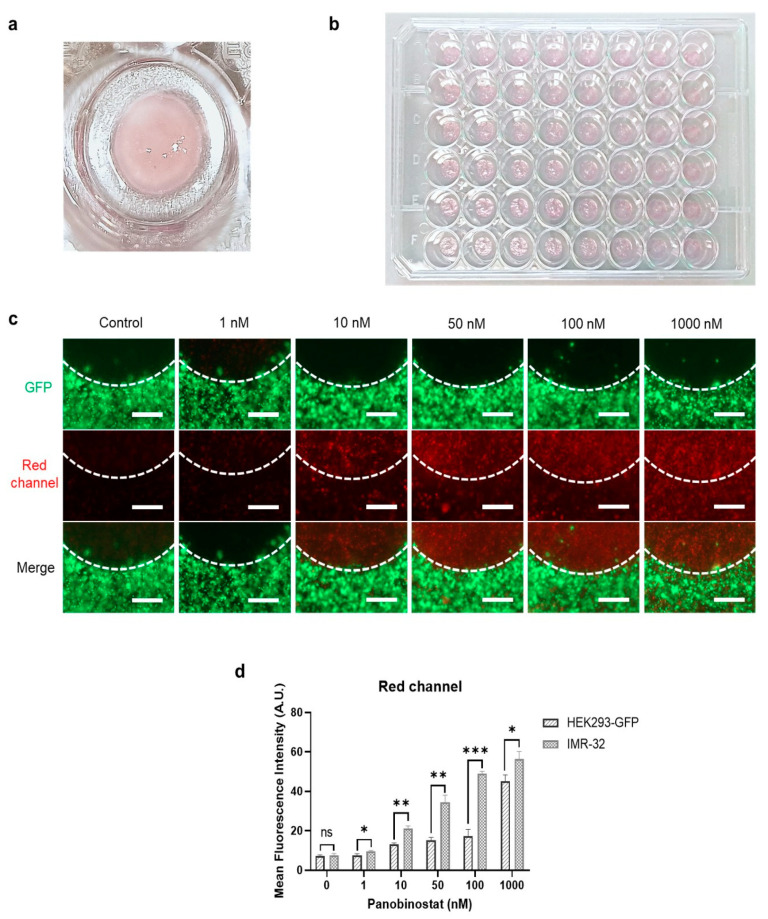
Cancer model and cytotoxicity assays following treatment with panobinostat. (**a**) The bioprinted cancer model consisted of a core of neuroblastoma IMR-32 cells (diameter of 3 mm, height of the construct 0.4 mm) surrounded by an environment of kidney cells (HEK293-GFP cells in these experiments, HEK293 cells and primary kidney fibroblasts, respectively, in experiments below, outer diameter is 6 mm). (**b**) Bioprinting can produce the constructs in a highly reproducible manner. (**c**) The cancer model was treated with panobinostat for 72 h and subsequently stained with ethidium homodimer-1. HEK293-GFP cells emitted green fluorescence, dead cells were identified by ethidium homodimer-1 staining in red. The boundary between both cell types is indicated by a dotted line. All images were taken at the same magnification; scale bar, 500 μm. (**d**) Quantitative analysis of dead cells in HEK293-GFP and IMR-32 parts of the models by mean fluorescence intensity based on the red channel results. * *p* < 0.05, ** *p* < 0.01, *** *p* < 0.001.

**Figure 7 ijms-23-00122-f007:**
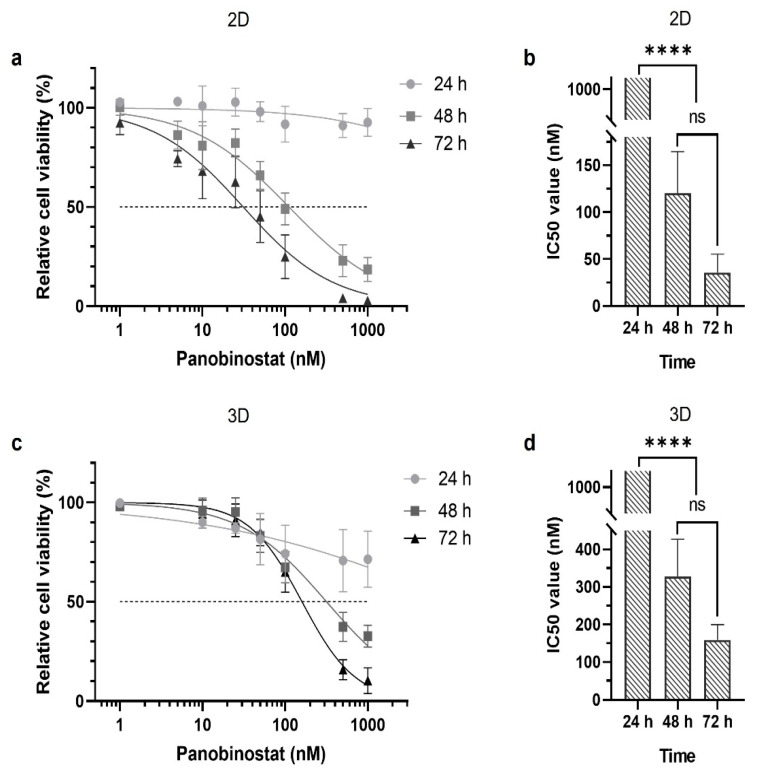
Dose–response curves for panobinostat treatment of primary human kidney fibroblasts in 2D and 3D culture. Monolayer (2D) cultured kidney fibroblast (**a**,**b**) and bioprinted (3D) kidney fibroblast (**c**,**d**) were treated with increasing concentrations of panobinostat. Cell viability was evaluated by XTT assays after 24, 48, and 72 h. Dose–response curves (**a**,**c**) and the calculated IC_50_ values (**b**,**d**) are shown. Data are presented as mean ± standard error of the mean; *n* = 3. **** *p* < 0.0001.

**Figure 8 ijms-23-00122-f008:**
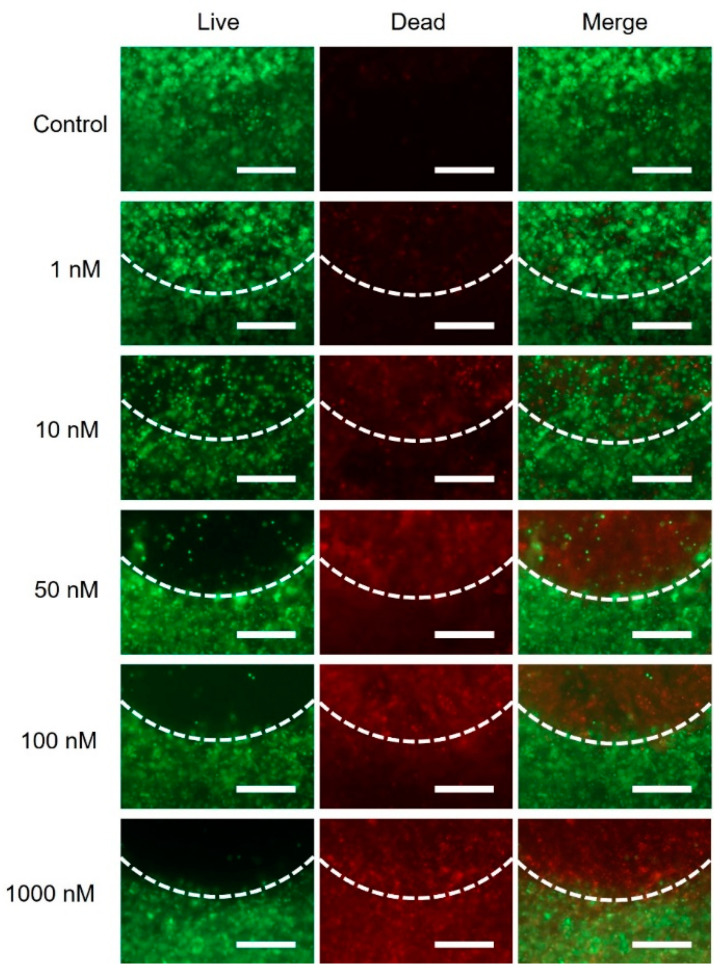
Cytotoxicity of cancer (IMR-32) and non-cancerous environment (primary kidney fibroblasts) of the bioprinted models after treatment with panobinostat. Cytotoxicity assays of cancer models were carried out after treatment with panobinostat for 72 h. Living cells were labeled in green, and dead cells were in red. The white dotted lines indicate the boundary between cancer part (above the line) and non-cancerous environment (below the line). All images were taken at the same magnification; scale bar, 500 μm.

**Table 1 ijms-23-00122-t001:** IC_50_ values of HEK293 and IMR-32 cells treated with panobinostat or blasticidin in 3D constructs or 2D culture.

		3D Bioprinted		2D Monolayer	
		HEK293	IMR-32	HEK293	IMR-32
Panobinostat (nM)	24 h	>1000	7.5 ± 0.1	>1000	4.9 ± 0.7
	48 h	509.0 ± 83.8	3.5 ± 0.5	40.5 ± 12.8	2.7 ± 0.5
	72 h	107.0 ± 40.1	2.1 ± 0.5	38.5 ± 7.8	2.5 ± 0.9
Blasticidin (µM)	24 h	44.3 ± 41.6	52.9 ± 15.2	7.9 ± 1.4	9.0 ± 2.8
	48 h	22.3 ± 9.1	21.9 ± 11.3	5.6 ± 0.1	5.2 ± 2.6
	72 h	15.1 + 6.5	12.4 ± 8.0	5.0 ± 1.1	3.9 ± 2.2

**Table 2 ijms-23-00122-t002:** IC_50_ values of human kidney fibroblast treated with panobinostat in 2D culture or 3D constructs.

		2D Monolayer	3D Bioprinted
Panobinostat (nM)	24 h	>1000	>1000
	48 h	120.3 ± 44.3	328.2 ± 98.5
	72 h	35.5 ± 19.8	158.9 ± 40.5

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
