# Peer review of "Bioprinted Cancer Model of Neuroblastoma in a Renal Microenvironment as an Efficiently Applicable Drug Testing Platform"

_ijms, 2021, doi:10.3390/ijms23010122_

Round 1

Reviewer 1 Report

The manuscript entitled “Bioprinted cancer model of neuroblastoma renal metastasis as an efficiently applicable drug testing platform” written by Dongwei Wu et al., has a rather satisfactory content, however there are some issues that should be clarified.

Major comments:

-  I suggest you to evaluate Bax and p53 expression to confirm the effect of Panobinostat on apoptosis pathway in HEK293 and IMR-32 cells.

- I suggest you to show the comparison among all groups in the figures (for example Figure 1A).

- I suggest you to ameliorate immunofluorescence (IF) image resolution and indicate the scale bar on the figures.

- Please give a reference for these sentences: “The TME has a major influence on the solid tumor, as it provides cytokines, immune cells, and vasculature that determine the tumor phenotype and encumber therapeutic interventions”; “The printability of neuroblastoma cells has been demonstrated in several studies”.

- It will be good if the Authors also provide a Figure for the proposed model of the study.

Minor comments:

- Please check the abbreviation because not all are defined at first mention.

- Specify the commercial brand, city and country of provenance for all materials used.

- The paper presents several errors; please take this opportunity to check better your manuscript for any typographical errors and to make any final correction or revisions.

Author Response

Reviewer 1

-  I suggest you to evaluate Bax and p53 expression to confirm the effect of Panobinostat on apoptosis pathway in HEK293 and IMR-32 cells.

In our study, we used cleaved caspase-3 as a marker for apoptosis. We agree that for a novel apoptosis inhibitor multiple assays should be performed. In this case, however, panobinostat was used for a proof-of-principle study that the model is suitable to test drugs. Reviewer 2 also remarks that panobinostat is already well-known as a trigger of apoptosis. Therefore, we did not investigate further apoptosis markers. In accordance with the suggestion by reviewer 2 (point 6), we rephrased two paragraphs (line 110 and line 218) to make clear that panobinostat was deliberately chosen as a well-characterized drug for proof-of-concept reasons, which we hope resolves this concern.

- I suggest you to show the comparison among all groups in the figures (for example Figure 1A).

As requested, all groups are now compared in one chart in the former Figure 1 (now Figure 2) as well as in the former Figure S1, which is now included as Figure 4 in the manuscript according to the request of reviewer 2.

- I suggest you to ameliorate immunofluorescence (IF) image resolution and indicate the scale bar on the figures.

The quality/resolution of all figures was improved for the final version of the manuscript and scale bars are now indicated on all fluorescence images as suggested.

- Please give a reference for these sentences: “The TME has a major influence on the solid tumor, as it provides cytokines, immune cells, and vasculature that determine the tumor phenotype and encumber therapeutic interventions”; “The printability of neuroblastoma cells has been demonstrated in several studies”.

Thank you, we added two references for the interaction of the TME with the tumor (references 46 and 47).

With regards to the printability of neuroblastoma cells, the sentence was meant as an introduction to the next sentences. We therefore use a colon at the end of the sentence now. References 30-35 in the following sentences summarize attempts to print neuroblastoma cells.

- It will be good if the Authors also provide a Figure for the proposed model of the study.

The new Figure 1 shows the models designed for this study.

Minor comments:

- Please check the abbreviation because not all are defined at first mention.

Thank you for this remark. We carefully checked the manuscript and now define all abbreviations at first mention.

- Specify the commercial brand, city and country of provenance for all materials used.

Thank you for this remark. We carefully checked the Materials and Methods section and now give the brand, city and country for all materials used.

The paper presents several errors; please take this opportunity to check better your manuscript for any typographical errors and to make any final correction or revisions.

Thank you, we underwent another round of careful proofreading by a native English-speaker. We think that all typographical errors have been corrected.

Reviewer 2 Report

In this manuscript, Dongwei et all reported a setting procedure to generate a 3D bioprinted cancer model for testing drugs, then used to compare the responsiveness of cancer and non-cancer cells to drugs according to alternative culture combinations. Specifically, the authors used human neuroblastoma IMR-32 cells either cultured alone or together with HEK-293 cells or primary human kidney fibroblasts. The authors proposed that this 3D coculture setting models neuroblastoma renal metastasis. Concerning inhibitors, the authors used the HDAC inhibitor panobinostat, showing that it preferentially targets cancer cells, and the peptidyl nucleoside antibiotic blasticidin, as a general cell death inducer of cancer and non-cancer cells. Two main outcomes are reported in this manuscript. First, the difference in the sensitivity of cancer and non-cancer cells to panobinostat, according to 2D and 3D culture settings: about two orders of magnitude when a 3D setting was used compared to an order of magnitude in 2D cultures. Second, the possibility to create a 3D model with cancer cells adjacent, or surrounded by, other cell types, to reconstitute a tumour microenvironment and to assess its influence on drug response.

Specific comments:

1) Through the text, the bioprinted 3D setting was presented by the authors in some parts as a model that “mimics a tumour in a microenvironment” and in other parts as a model “for metastasized neuroblastoma”. I consider the second definition too speculative, as uniquely based on the use of two cell types with distinct origins (neuroblastoma from the brain together with cells of kidney origin). No data are reported in relation to molecular and/or biological metastatic features. Therefore, the authors should appropriately rephrase in the manuscript the bioprinted 3D model they used.

2) I strongly recommend to add a scheme in Figure 1 to illustrate the 3D system used, with the different experimental settings employed in the study, to guide readers to appreciate differences. Additionally, I recommend to add a clear image illustrating the 3D culture, for example with DAPI staining (or DAPI plus GFP to show the two cell types), and ideally a 3D image reconstitution.

3) The 3D co-cultures used by the authors (reported in Figure 4, 5, and 6) were based on the use of neuroblastoma cells within the core, surrounded by cells of kidney origin. The authors must use as well the reverse setting (kidney cells in the core with neuroblastoma cells in the surrounding area) and dose response curves to panobinostat treatment in these two settings should be compared. This comparison is important to exclude additive effects linked, for example, to limited accessibility of nutrients for cells within the core compared to those in the periphery of the aggregates. Importantly, this setting will serve as a reference for future studies using other parameters (drugs, cell types, …).

4) I recommend to include Figure S1 in the manuscript.

5) In the legend of Figure S2 and S6, the authors stated that: "Living cells were stained in green while dead cells were stained in red, following image capture with a fluorescence microscopy". However, no red staining is visible in the figure, at least with the quality of the images I had access.

6) In the result section 2.4, the author stated that "The next aim was to study the mode of action of panobinostat that is known to be an HDAC inhibitor". However, several published reports already established that panobinostat triggers apoptosis. Therefore, it is not clear which is really the relevance of this aim for lack of novelty. Consequently, these data should be presented in the study more properly.

7) In Figure S3, cleaved Caspase3 staining is barely visible. I strongly recommend to increase the contrast or show better images.

8) Death of cells in experiments reported in Figure 4 and Figure S4 should be quantified, to allow direct comparison with those reported in Figure 1 and 2.

9) Among advantages for the use of the 3D system reported in this manuscript, I suggest to the authors to discuss as well the possibility to limit the number of animals used for experimental studies (the 3Rs principles).

10) In line 300 (page 9), there is a message "Error! Reference source not found", which needs to be corrected (supposedly Figure 6).

11) From line 403 to 406, there is a problem of formatting style.

Author Response

Reviewer 2

 Specific comments:

1) Through the text, the bioprinted 3D setting was presented by the authors in some parts as a model that “mimics a tumour in a microenvironment” and in other parts as a model “for metastasized neuroblastoma”. I consider the second definition too speculative, as uniquely based on the use of two cell types with distinct origins (neuroblastoma from the brain together with cells of kidney origin). No data are reported in relation to molecular and/or biological metastatic features. Therefore, the authors should appropriately rephrase in the manuscript the bioprinted 3D model they used.

We followed the advice to use a more modest phrasing. To this end, we no longer use the term metastasized neuroblastoma for our tumor model. Changes were made throughout the text, and we also adjusted the title. In line 89, we state that this model can be regarded as a simplified metastases model to underline our motivation for the design of our model.

2) I strongly recommend to add a scheme in Figure 1 to illustrate the 3D system used, with the different experimental settings employed in the study, to guide readers to appreciate differences. Additionally, I recommend to add a clear image illustrating the 3D culture, for example with DAPI staining (or DAPI plus GFP to show the two cell types), and ideally a 3D image reconstitution.

We included a new Figure 1 with 3D images of the constructs used in the present study as requested. With regard to the DAPI staining, Figure S4 shows a respective stain of the whole construct.

3) The 3D co-cultures used by the authors (reported in Figure 4, 5, and 6) were based on the use of neuroblastoma cells within the core, surrounded by cells of kidney origin. The authors must use as well the reverse setting (kidney cells in the core with neuroblastoma cells in the surrounding area) and dose response curves to panobinostat treatment in these two settings should be compared. This comparison is important to exclude additive effects linked, for example, to limited accessibility of nutrients for cells within the core compared to those in the periphery of the aggregates. Importantly, this setting will serve as a reference for future studies using other parameters (drugs, cell types, …).

While the reviewer raises an interesting point, we do not think that this additional control to produce a model with kidney cells in the core surrounded by neuroblastoma cells is necessary. As can be seen in Figures 3 and 6, most of the cells are viable in the control. For example, in the control of Figure 6 strong green fluorescence indicating viable cells, but red fluorescence representing dead cells is extremely weak. This is the same case in both the cancer part (center) and normal part (surrounding). This clearly confirms that all cells have similar and sufficient supply with nutrients etc. There is obviously no shortcoming in the center. In addition, differing results in the inverse conformation cannot be simply interpreted as a failure of the control due to limited access to nutrients or some other factors, and it would trigger an extensive set of experiments to understand the difference. The short time frame made it impossible for us to consider carrying out this experiment in hopes that the inverse conformation provides the expected results.

4) I recommend to include Figure S1 in the manuscript.

Figure S1 is included now in the main manuscript as Figure 4 as requested.

5) In the legend of Figure S2 and S6, the authors stated that: "Living cells were stained in green while dead cells were stained in red, following image capture with a fluorescence microscopy". However, no red staining is visible in the figure, at least with the quality of the images I had access.

We have improved the quality of the figures so that the red cells become more visible now (it is easier to see them when magnifying the figure). For the 2D cultures, it should be noted that most of the dead cells detached from the culture plate and were removed during the staining procedure.

6) In the result section 2.4, the author stated that "The next aim was to study the mode of action of panobinostat that is known to be an HDAC inhibitor". However, several published reports already established that panobinostat triggers apoptosis. Therefore, it is not clear which is really the relevance of this aim for lack of novelty. Consequently, these data should be presented in the study more properly.

We fully agree that panobinostat is a well-established, approved, and well-characterized substance. It was not our aim to present it as a new anti-cancer drug. Rather, we wanted to use it for a proof-of-concept that our model can be used to study the effect of potential inhibitors of tumor growth. We therefore followed the reviewer’s suggestion and made some changes to present the findings more properly. On line 218, we replaced the Phase: “The next aim was to study the mode of action of panobinostat” by “…confirm observation of this mode of action in our experimental set-up.…”. Furthermore, we wrote that panobinostat was chosen for proof-of-principle studies on line 110. These changes also refer to the first remark of reviewer 1.

7) In Figure S3, cleaved Caspase3 staining is barely visible. I strongly recommend to increase the contrast or show better images.

Thank you for the advice. We improved the quality of the figure as requested.

8) Death of cells in experiments reported in Figure 4 and Figure S4 should be quantified, to allow direct comparison with those reported in Figure 1 and 2.

As suggested by the reviewer, we included a quantitative analysis in the figures (now Figures 6 and S3) and added a sentence to the text (line 249).

9) Among advantages for the use of the 3D system reported in this manuscript, I suggest to the authors to discuss as well the possibility to limit the number of animals used for experimental studies (the 3Rs principles).

We are extremely grateful for this remark. As the authors of this study are heavily involved in various 3R activities (the corresponding author JK, for example, is co-speaker of the newly founded Einstein Center 3R in Berlin), the replacement of animal experiments is one of the main motivations of our research. We did not strongly emphasize this point in the original manuscript since the work must be able to stand on its own without the need for additional justification through its potential with respect to 3R. However, after reading this remark, we realized we may not have given 3R enough weight, so we added a short paragraph and an additional reference to the end of the discussion section, in which we state that the aims to provide better models for human (patho-)physiology and to reduce the number of animal experiments are closely connected.

10) In line 300 (page 9), there is a message "Error! Reference source not found", which needs to be corrected (supposedly Figure 6).

Thank you, we corrected this error (it was Figure 6 and is now Figure 8)

11) From line 403 to 406, there is a problem of formatting style.

Thank you for calling this to our attention. We corrected the font.